# Light and Shadows: Insights from Large-Scale Visual Screens for Arabidopsis Leaf Morphology Mutants

**DOI:** 10.3390/ijms26178332

**Published:** 2025-08-28

**Authors:** Lucía Juan-Vicente, Alejandro Ruiz-Bayón, José Luis Micol

**Affiliations:** Instituto de Bioingeniería, Universidad Miguel Hernández, Campus de Elche, 03202 Elche, Spain; ljuan@umh.es (L.J.-V.); alejandro.ruizb@umh.es (A.R.-B.)

**Keywords:** forward genetic screens, mutant collection, plant leaf morphogenesis

## Abstract

Screens for specific phenotypes have long been a cornerstone of biology. Here, we present an updated synthesis of our large-scale visual screens for Arabidopsis (*Arabidopsis thaliana*) mutants that exhibit leaf morphology defects. In our 2009 review, we used phenotypes to group the leaf mutants that we had isolated and characterized since 1992; here, by contrast, we functionally classified the mutations that we studied over the last 16 years based on the biological programs they disrupt. Since 2009, we have identified and analyzed 38 genes required for proper leaf development; these genes are involved in translation, chloroplast function, cell wall construction, auxin homeostasis, microRNA biogenesis, and epigenetic regulation. Many of the identified mutants have pleiotropic phenotypes, consistent with the central roles of the affected pathways in development. In this review, we systematically link morphological traits to specific molecular dysfunctions, highlighting the enduring utility of forward genetic approaches. We found that the Arabidopsis leaf is a model organ of a model organism, and we have used this model-in-a-model system to dissect whole-plant traits such as cell proliferation and expansion, and to improve our understanding of the genetic control of plant form and size.

## 1. Introduction

Mutants, particularly from curated mutant collections, have long served as powerful tools for uncovering the genetic and molecular mechanisms underlying biological principles [1]. The first systematic mutant collections were established in the fruit fly *Drosophila melanogaster* by identifying mutations that altered wing shape or wing venation pattern [2]. *D. melanogaster* also offers a paradigmatic example of the usefulness of mutant collections: the mutants affecting embryonic development obtained by Nüsslein-Volhard and Wieschaus [3]. Other examples of collections of mutants in a model species include those affecting vulval differentiation in the nematode *Caenorhabditis elegans* [4] and those exhibiting diverse phenotypes in the mouse (*Mus musculus*) [5,6], which led to the identification of genes involved in fertility [7] and clonal B-cell lymphomagenesis [8].

In the plant kingdom, maize (*Zea mays*) and Arabidopsis (*Arabidopsis thaliana*) are prime examples of how mutant analysis has advanced our understanding of plant biology. Our research focuses on Arabidopsis, which was proposed as a model organism by Friedrich Laibach in 1943 [9] and has become a pillar of plant genetics research. Arabidopsis mutant collections generated through chemical, physical, and insertional mutagenesis have helped researchers elucidate diverse phenomena, including embryo development [10,11,12] and leaf morphogenesis [13]. These efforts have deepened our understanding of plant development and yielded an invaluable set of genetic resources for the plant science community [14].

The collections mentioned above primarily represent forward genetic approaches, in which mutagenesis was followed by screening to identify mutants with a phenotype of interest and characterization of the underlying genetic alterations. By contrast, reverse genetic approaches involve generating collections of mutants with altered expression of individual genes, allowing researchers to assess the phenotypes associated with known, and in most cases indexed, gene disruptions. Examples of reverse genetics resources include RNA interference (RNAi) lines targeting 71% of all genes in *D. melanogaster* [15], lines obtained using clustered regularly interspaced short palindromic repeats (CRISPR)/CRISPR-associated protein (Cas) for gene knockout or gene overexpression also in *D. melanogaster* [16], mutants obtained by transposon-based mutagenesis in *C. elegans* [17], and a comprehensive knockout collection in mice [18]. In Arabidopsis, two landmark collections of T-DNA insertion lines have enabled systematic functional studies across the genome: the SALK lines [19] and the Syngenta Arabidopsis Insertion Library (SAIL) lines [20].

Although mutants are invaluable for identifying genes involved in specific aspects of biology, characterization of mutants often reflects the research focus of the laboratories analyzing them. For example, David W. Meinke, who pioneered the study of *embryo-defective* (*emb*) mutants in Arabidopsis, noted that researchers who focused on gametophyte development frequently overlooked embryonic phenotypes. Conversely, Meinke and his collaborators generally excluded mutations that caused gametophytic lethality. He also highlighted a bias introduced by the prevalence of reverse genetic studies: the overrepresentation of *emb* mutants corresponding to well-studied protein families, reflecting research trends rather than biological importance [21,22].

Forward genetics helps circumvent researcher biases and enables the discovery of unexpected gene functions. For example, *FLOWERING CONTROL LOCUS A* (*FCA*) encodes an RNA-binding protein involved in post-transcriptional regulation; its effect on flowering time was discovered by a forward genetic approach [23]. The outcomes of forward genetic approaches, however, are not always aligned with the expectations of the researchers. For example, in our laboratory, screening for mutants with altered leaf morphology has led to the identification of genes affecting leaf morphogenesis as well as fundamental developmental processes throughout the Arabidopsis body and life cycle (Appendix A). This observation mirrors the findings of Jürgens and colleagues, whose screens for embryonic patterning mutants also uncovered alleles of genes controlling essential cellular functions [11], rather than exclusively identifying specialized regulators of early development.

Over the years, we have conducted three large-scale screens for leaf morphological mutants in Arabidopsis: one using ethyl methanesulfonate (EMS) treatment [13], one using fast-neutron (FN) bombardment [24], and a third selecting publicly available lines from The Arabidopsis Information Service (AIS) collection [25]. To expand our collection, we also screened the SALK T-DNA insertion lines, identifying 706 additional leaf mutants, which we cataloged in our PhenoLeaf database [26].

In recent years, several reviews have been published on different facets of leaf development [27,28,29,30,31,32,33,34,35,36,37,38,39,40,41,42,43,44,45,46,47,48,49]. Some of these examine leaf organogenesis through the lens of mechanics [30,34,37,40] or evolution [39,45], and others discuss the contribution of ribosomal proteins [42] and hormones [41,48], or focus on genetics [47] or development of the leaf epidermis [36,49]. In the present review, we update and functionally reinterpret the collection of Arabidopsis leaf morphology mutants isolated in our laboratory, and characterized by our group and others since our previous synthesis in 2009 [50], incorporating the last 16 years of our research. In contrast to our previous phenotypic classification, we now organize our mutants based on the biological programs that their causal mutations disrupt, thereby aiming to provide a clearer view of gene function. We also highlight one exceptional mutant from the PhenoLeaf collection [26], notable for its unusual loss of leaf bilateral symmetry, and present a comprehensive table summarizing all characterized and pending mutant lines. In addition, we compiled the results of all double mutant combinations generated and published by our group since 1992, whose synergistic or additive morphological phenotypes either reveal or allow us to exclude putative genetic interactions. These results help define the functions of individual genes and the functional relationships within and between developmental pathways. In Section 2.1, Section 2.2, Section 2.3, Section 2.4, Section 2.5, Section 2.6 and Section 3, we examine the functional categories of our leaf morphology mutants, i.e., translation, chloroplast biogenesis and function, cell wall biosynthesis, auxin homeostasis, miRNA biogenesis and function, and epigenetic regulation. These categories, spanning several major processes of plant cells, give an indication of the extent of the insights provided by studying mutants exhibiting altered leaf morphology. All the mutations we have obtained and will discuss in this review are loss-of-function mutations, except where explicitly stated otherwise.

## 2. Leaf Morphology Mutants Studied Since 2009

### 2.1. Leaf Morphology Mutants with Defects in Translation

At least 18 mutants from our whole EMS collection (~9%) carry mutations in genes associated with translation; many of these mutations damage genes encoding ribosomal proteins (Appendix A). Although ribosomal proteins have been named based on whether they belong to the large subunit or the small subunit (RPLs and RPSs, respectively), an updated and consistent cross-kingdom nomenclature has been proposed [51,52]. However, here we use their originally published names for the sake of simplicity and provide their updated names in Appendix A.

The first plant mutant affecting a gene encoding a ribosomal protein (RPS18A) was Arabidopsis *pointed first leaves* (*pfl*), which was isolated in the laboratory of Mieke Van Lijsebettens [53]. As its name suggests, the phenotype of *pfl* is most apparent on the first true leaves of seedlings, which are pointed instead of rounded [53]. Our *angusta3* (*ang3*) mutant, which carries a mutation in *RPL5B* (*uL18y*), displays narrower leaves with a lower trichome density, smaller abaxial epidermal cells, and shorter hypocotyls and primary roots compared to the wild type (Figure 1B and Appendix A; ref. [54]). Our *apiculata2* (*api2*) mutant harbors a mutation in *RPL36aB* (*eL42y*), has pointed leaves and a main stem with reduced height (Figure 1C and Appendix A; ref. [55]).

The mutants we initially named *denticulata5* (*den5*), *den12*, and *den30* carry mutations in *RPL7B* (*uL30y*), *RPL10aB* (*uL1y*), and *RPL39C* (*eL39x*), respectively, and were thus renamed *rpl7b-1*, *rpl10ab-3*, and *rpl39c-1*, respectively. Additionally, the *den29* mutant carries a deletion of two adjacent genes, *RPS15aB* (*uS8mz*) and *RPL28A* (*eL28z*), and was renamed as the *rps15ab-1 rpl28a-3* double mutant. All *den* mutants have small, reticulated, narrow and pointed leaves, with teeth visible on their first- and second-node leaves (Figure 1E–I and Appendix A). The palisade mesophyll cells of these mutants are larger than those of the wild type and display increased ploidy levels. Several *den* mutants have additional distinct traits: the leaves of the *den5* and *den29* mutants show parallel veins on their proximal lamina, and *den29* and *den30* show a disorganized palisade mesophyll [56]. The *den2* mutant has pointed, toothed leaves (Figure 1E and Appendix A), and carries a mutation in *SMALL ORGAN4* (*SMO4*), which encodes a ribosome biogenesis factor involved in 5.8S rRNA maturation; it was characterized in the laboratory of María Rosa Ponce [57]. We are currently studying the *den3*, *den4*, *den6*, *den7*, *den9*, *den13*, *den14*, and *den15* mutants (Appendix A), all of which exhibit a similar phenotype; most, but not all, of these mutants carry mutations in genes related to the translational apparatus.

**Figure 1 ijms-26-08332-f001:**
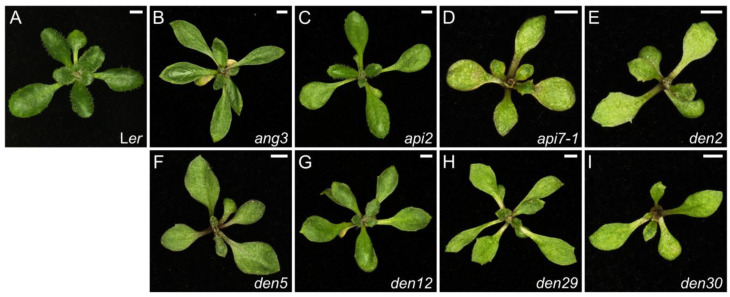
Leaf phenotypes of mutants with defects in translation. Rosettes of the wild-type Landsberg *erecta* (L*er*) (**A**), and the homozygous mutants *ang3* (**B**), *api2* (**C**), *api7-1* (**D**), *den2* (**E**), *den5* (*rpl7b-1*) (**F**), *den12* (*rpl10ab-3*) (**G**), *den29* (*rps15ab-1 rpl28a-3*) (**H**), and *den30* (*rpl39c-1*) (**I**). Photographs were taken 21 days after stratification (das). Scale bars, 2 mm. These mutants were described in [54] (**B**), ref. [55] (**C**), ref. [58] (**D**), ref. [57] (**E**), and [56] (**F**–**I**). All figures in this review feature photographs taken concurrently from plants cultivated in parallel under the same controlled growth conditions.

The *apiculata7-1* (*api7-1*) mutant carries a mutation in *ATP-BINDING CASSETTE E2* (*ABCE2*), which encodes a protein responsible for dissociating cytoplasmic ribosomes at the end of translation and thus contributes to ribosome recycling. The *api7-1* plants exhibit a phenotype typical of ribosomal protein mutants, namely a small rosette and pointed, toothed, and pale leaves that contain reduced levels of photosynthetic pigments and have a simple venation pattern (Figure 1D and Appendix A). Additionally, both the leaf epidermal cells and the overall size of *api7-1* plants are reduced compared to the wild type [58].

All the mutants affected in genes encoding components of the translational apparatus that we isolated in our screens share a common phenotype characterized by pointed, narrow, and toothed leaves, along with alterations in cell size across all leaf layers. However, why mutations affecting translation cause this phenotype remains unclear. Reduced protein synthesis can explain the diminished or retarded growth of these mutants, as was suggested long ago for the Minute phenotype of *Drosophila melanogaster*, which includes prolonged development and short, thin bristles. Indeed, the connection between the *Minute* mutants and defects in protein translation was first proposed in the late 1970s and early 1980s, when researchers began discovering that many *Minute* genes encode ribosomal proteins [59,60].

The narrow-leaf phenotype of the *den* mutants may reflect the role of ribosomal proteins in regulating the translation of transcripts that harbor upstream open reading frames (uORFs). For example, in rice (*Oryza sativa*), loss of OsRPS3A compromises the reinitiation of translation in uORF-containing mRNAs such as those transcribed from *AUXIN RESPONSE FACTOR 11* (*OsARF11*), *OsARF16*, and *WUSCHEL-RELATED HOMEOBOX* (*OsWOX3A*), ultimately leading to a narrow-leaf phenotype [61]. Indeed, despite their shared traits, the phenotypes of our *den* mutants differ, possibly due to the depletion of specific ribosomal proteins that may differentially affect the translation of certain transcripts, leading to distinct phenotypes [56].

In Arabidopsis, 249 genes encode the 81 distinct proteins that constitute the cytosolic ribosome, reflecting a remarkable heterogeneity compared to animals, whose genomes usually harbor fewer than 100 ribosomal protein genes. One plausible explanation for the large number of ribosome protein genes in plants is that highly proliferative tissues, such as leaf primordia, may require elevated expression of ribosomal protein genes. The correlation between differential expression of ribosomal protein paralogs and cell division rates suggests that the phenotypic differences that we observed may result from dosage effects rather than functional specialization [42,62,63].

### 2.2. Leaf Morphology Mutants with Defects in Chloroplast Biogenesis and Function

The chloroplast consists of thylakoids organized in grana and surrounded by stroma, all enclosed by two membrane layers. Many chloroplast proteins are encoded by nuclear genes and chloroplasts affect nuclear gene expression by retrograde signaling [64]. We identified 27 mutants (~13%) in our screens with mutations in genes related to chloroplast function (Appendix A). We used next-generation sequencing to identify the causal mutations of the phenotypes of four *angulata* (*anu*) mutants with pale-yellow leaf laminae and irregular leaf margins with prominent teeth (*anu1-1*, *anu4-1*, *anu9-1*, and *anu12-1*; Figure 2B,C,E,G and Appendix A; ref. [65]). The *anu1-1* and *anu4-1* mutants were found to harbor mutations in the *SECA2* and *TRANSLOCON AT THE OUTER MEMBRANE OF CHLOROPLASTS33* (*TOC33*) genes, respectively. The secretory pathway SECA2 protein, together with the TOC33 protein, participate in the import of proteins from the cytosol into chloroplasts [66,67]. *NON-INTRINSIC ABC PROTEIN14* (*NAP14*; [68]) encodes a protein that is truncated by the *anu9-1* mutation and has a role in chloroplast homeostasis of transition metals such as iron and manganese, which are essential for photosynthesis [69]. The *anu12-1* mutation damages the coding sequence of *CLP PROTEASE PROTEOLYTIC SUBUNIT1* (*CLPR1*), whose product is involved in protein degradation in chloroplasts [70].

The two remaining *anu* mutants with chloroplast aberrations identified in our screens carry mutations that disrupt genes with functions that remain to be elucidated. The *anu7-1* and *anu10-1* plants exhibit leaves that are toothed, pale green, reduced in length, and display variation in the size of palisade mesophyll cells, as well as low levels of chlorophylls and carotenoids (Figure 2D,F and Appendix A; ref. [71,72]). In addition, the *anu10-1* mutant is compromised in leaf lateral expansion and accumulates H_2_O_2_ in its chloroplasts, which are small and abnormally shaped with fewer thylakoid membranes per granum [71,72]. ANU10 is present in the thylakoid membrane and is presumably required for thylakoid biogenesis and grana stacking [71]. ANU7 has a conserved DnaJ-like cysteine-rich domain, and other DnaJ proteins participate in various processes that take place in chloroplasts, like maintenance of photosystem II under chilling stress or chloroplast iron–sulfur cluster biogenesis [72,73,74]. Furthermore, we combined the *anu7-1* mutation with a loss-of-function allele of *GENOMES UNCOUPLED 1* (*GUN1*), which encodes a chloroplast-localized pentatricopeptide repeat (PPR) protein with a C-terminal small MutS-related domain. GUN1 plays a central role in chloroplast-to-nucleus retrograde signaling, but its specific function remains controversial, although it apparently involves the maturation of plastid transcripts [75,76,77]. The *anu7-1 gun1* double mutant (Appendix A) shows a synergistic phenotype with albino to completely green leaf sectors, thus revealing a genetic interaction [72].

The *rugosa1* (*rug1*) plants are shorter and flower later than the wild type; they also produce irregularly shaped leaves that develop lesions (Figure 2K and Appendix A). The *rug1* mutation damages the gene that encodes HYDROXYMETHYLBILANE SYNTHASE (HEMC, also reported as PORPHOBILINOGEN DEAMINASE [PBGD]), which is involved in tetrapyrrole biosynthesis, and therefore, in chlorophyll biosynthesis [78]. The *hemc-1* mutant exhibits altered RNA editing efficiency of four chloroplast transcripts [79]. Since chloroplast RNA editing and retrograde signaling appear to be somewhat related [80], RUG1 may also have a role in retrograde signaling, but this has not yet been experimentally confirmed. The *rug2* mutant has pale and small hypocotyls, rosettes, stems and fruits, and short roots, as well as leaves with green and pale sectors, the latter containing sparsely packed mesophyll cells with chloroplasts abnormally shaped and reduced in number compared to wild-type cells (Figure 2L and Appendix A; ref. [81]). *RUG2* is also known as *BELAYA SMERT* (*BSM*) and encodes a homolog of metazoan mitochondrial transcription termination factor (mTERF4). Arabidopsis mTERF4 has a role in intron splicing in chloroplasts [82,83].

*VENOSA3* (*VEN3*) and *VEN6* encode the large and small subunits, respectively, of carbamoyl phosphate synthetase (CPS) [84]. This enzyme is located in chloroplasts and generates carbamoyl phosphate, which is required for arginine and pyrimidine biosynthesis [85,86]. The *ven3* and *ven6* mutants have reticulated leaves with green veins and a generally pale lamina; their juvenile leaves are hyponastic, and adult leaves are more toothed than those of the wild type (Figure 2M–P,U and Appendix A). These reticulated leaves have fewer palisade mesophyll cells [84]. We discovered that *VEN3* and *VEN6* are overexpressed in the *orbiculata1-3* (*orb1-3*) mutant. *ORB1* encodes the chloroplast-localized GLUTAMATE SYNTHASE1 (GLU1), which is also referred to as FERREDOXIN-DEPENDENT GLUTAMINE OXOGLUTARATE AMINOTRANSFERASE1 (FD-GOGAT1) [87,88,89]. The leaf lamina of *orb1* mutants is smaller and exhibits diminished growth caused by small palisade mesophyll cells, and pale green pigmentation due to low levels of chlorophyll *a*, chlorophyll *b*, and carotenoids (Figure 2H–J and Appendix A). As CPS and GLU1 both use glutamine as their substrate, the overexpression of *VEN3* and *VEN6* in the *orb1-3* mutant may be a compensatory mechanism to lower its high glutamine levels [90].

Another mutant with reticulated and pale leaves is *ven4-0* (Figure 2Q and Appendix A), which carries a point mutation in *VEN4*, encoding the most likely Arabidopsis ortholog of human Sterile alpha motif and histidine-aspartate domain containing protein 1 (SAMHD1). Like human SAMHD1, Arabidopsis VEN4 is a deoxynucleotide triphosphohydrolase (dNTPase) involved in maintaining the dNTP pool [91] and in repairing DNA double-stranded breaks by homologous recombination [92]. The *ven4-0* mutant also accumulates lower levels of chlorophyll and has smaller chloroplasts in palisade mesophyll cells, with more plastoglobuli and poorly organized thylakoids [91]. This lower chloroplast number may be a consequence of insufficient chloroplast genome replication due to lower dNTP levels, a trait also observed in the *crinkled leaves 8-1* (*cls8-1*) mutant, which carries a mutation in *CLS8* encoding a protein involved in de novo dNTP biosynthesis [93]. *VEN5*, also known as *RETICULATA-RELATED3* (*RER3*), encodes a protein of unknown function localized in the chloroplast envelope. We renamed our *ven5-1*, *ven5-2*, and *ven5-3* mutants, all of which carry point mutations in *RER3*, to *rer3-1*, *rer3-2*, and *rer3-3*, respectively. These mutants exhibit reticulated, hyponastic, and toothed leaves (Figure 2R–T and Appendix A). Additionally, the leaf cells of these mutants show lower ploidy levels, their epidermal cells are smaller and have a disorganized mesophyll structure. Accumulation of H_2_O_2_ in the tissue surrounding their vasculature leads to localized cell death and causes leaf reticulation [94].

*SCABRA1* (*SCA1*) encodes PLASTID RIBOSOMAL PROTEINS5 (RPS5). The *sca1-1* and *sca1-2* plants are smaller than the wild type, produce pale green leaves with lower contents of chlorophyll *a* and *b* and lower maximum efficiency of photosystem II than the wild type, together with an uneven leaf surface, prominent marginal teeth, and small palisade mesophyll cells (Figure 2V,X and Appendix A). The *sca1-1* mutation enhances the phenotype observed for *asymmetric leaves2-1* (*as2-1*; ref. [95]), as shown by the leaf polarity defects of the *sca1-1 as2-1* double mutant (Appendix A), which produced radialized leaves and some trumpet-shaped leaves, uncovering a role for plastids in the establishment of adaxial–abaxial polarity [96]. As mentioned above, the *den10* and *den17* mutants are currently under study in our laboratory and carry mutations in genes that encode chloroplast components (Appendix A).

**Figure 2 ijms-26-08332-f002:**
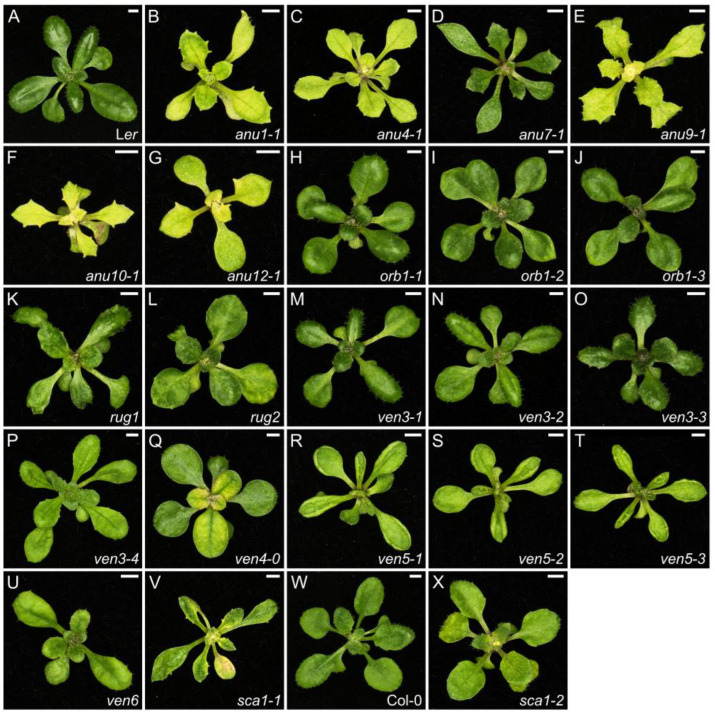
Leaf phenotypes of mutants with altered chloroplast biogenesis and function. Rosettes of the wild-types L*er* (**A**) and Columbia (Col-0) (**W**), and the *anu1-1* (**B**), *anu4-1* (**C**), *anu7-1* (**D**), *anu9-1* (**E**), *anu10-1* (**F**), *anu12-1* (**G**), *orb1-1* (**H**), *orb1-2* (**I**), *orb1-3* (**J**), *rug1* (**K**), *rug2* (**L**), *ven3-1* (**M**), *ven3-2* (**N**), *ven3-3* (**O**), *ven3-4* (**P**), *ven4-0* (**Q**), *ven5-1* (*rer3-1*) (**R**), *ven5-2* (*rer3-2*) (**S**), *ven5-3* (*rer3-3*) (**T**), *ven6* (**U**), *sca1-1* (**V**), and *sca1-2* (**X**) homozygous mutants. Photographs were taken 21 das. Scale bars, 2 mm. These mutants were described in [65] (**B**,**C**), ref. [72] (**D**), ref. [65] (**E**), ref. [71] (**F**), ref. [65] (**G**), ref. [90] (**H**–**J**), ref. [78] (**K**), ref. [81] (**L**), ref. [84] (**M**–**P**), ref. [91] (**Q**), ref. [94] (**R**–**T**), ref. [84] (**U**), ref. [96] (**V**), and [96] (**X**).

Not unexpectedly, mutations in genes associated with the biosynthesis of photosynthetic pigments and chloroplast structure affect leaf morphology. The lower accumulation of photosynthetic pigments results in pale or reticulated leaves, whereas abnormalities in chloroplast structure and function impair cell expansion, altering the final leaf shape and size, leading, in this case, to toothed leaf margins and smaller rosettes.

### 2.3. Leaf Morphology Mutants Altered in Cell Wall Biosynthesis

Organ size and shape in plants are governed by cell proliferation and expansion, both of which depend on precise control of cell wall biogenesis. Cell division involves the formation of new cell walls at the division plane and cell expansion requires the insertion of newly synthesized polysaccharides to enable wall loosening and elongation [97]. Therefore, mutations that disrupt any aspect of cell wall construction may directly influence leaf shape. From our EMS mutagenesis, we have identified seven mutants (~3%) affected in genes related to cell wall construction (Appendix A).

Our *serrata4-1* (*sea4-1*) and *sea4-2* mutations are the only known viable mutant alleles of *KEULE* (*KEU*), which encodes a Sec1/Munc18 (SM) protein that interacts with the syntaxin KNOLLE (KN) in the plane of division [98]. This interaction facilitates the formation of *trans*-SNARE [soluble N-ethylmaleimide-sensitive factor (NSF)-attachment protein receptor] complexes between vesicles that then fuse to form the new cell wall between the two daughter cells during cytokinesis [99]. The *sea4-1* and *sea4-2* plants have small rosettes, with fewer leaves at bolting compared to wild type, as well as shorter stems and roots. Their leaves are serrated, small, and wavy, displaying a more complex venation pattern than the wild type, and undergoing premature senescence (Figure 3G,H and Appendix A). Their leaf mesophyll cells are generally smaller, but epidermal cells have a simpler shape with fewer protuberances, and both mutants show increased levels of endoreduplication. The transcriptomic changes observed in the *sea4* mutants suggest a drop in cell wall integrity, which may contribute to the observed phenotype [100].

The *exigua1* (*exi1*), *exi2*, and *exi5* mutants carry point mutations in the *CELLULOSE SYNTHASE A8* (*CESA8*), *CESA4*, and *CESA7* genes, respectively. These genes encode enzymes that synthesize cellulose microfibrils in the secondary cell wall during its formation [101]. These *exi* mutants exhibit small, dark-green leaves, short stems, roots and siliques, and small flowers (Figure 3B–E and Appendix A). Additionally, their leaf cells are smaller and have a lower ploidy level compared to wild type, as also observed in other *cesa* mutants such as *cesa6* [102]. In the *exi1*, *exi2*, and *exi5* mutants, CESA dysfunction appears to cause the collapse of xylem vessels, which in turn reduces the turgor pressure necessary for proper cell expansion [103].

**Figure 3 ijms-26-08332-f003:**
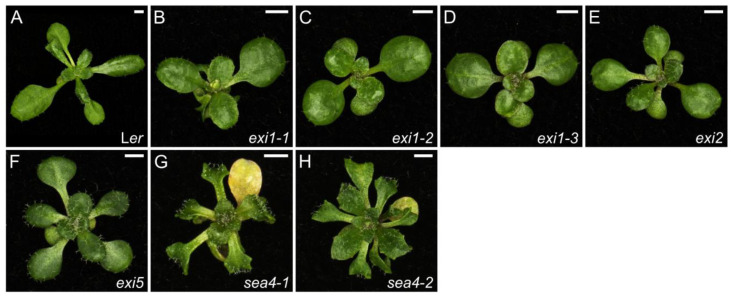
Leaf phenotypes of mutants altered in cell wall biosynthesis. Rosettes of the wild-type L*er* (**A**), and the *exi1-1* (**B**), *exi1-2* (**C**), *exi1-3* (**D**), *exi2* (**E**), *exi5* (**F**), *sea4-1* (**G**), and *sea4-2* (**H**) homozygous mutants. Photographs were taken 21 das. Scale bars, 2 mm. These mutants were described in [103] (**B**–**F**), and [100] (**G**,**H**).

Due to the importance of cell wall construction for the complete development of the plant body, mutations in genes involved in cell wall biosynthesis result in a pleiotropic morphological phenotype. Indeed, the *sea4*, *exi1*, *exi2*, and *exi5* mutants exhibit alterations in rosette size, stem and root length, leaf shape, color, size, venation pattern and internal structure, inflorescences, and ploidy. Cytokinesis is required for leaf growth as shown in *sea4* mutants. In addition, those mutants have lost cell wall integrity, which triggers an immune response followed by senescence. Defective cell wall construction also compromises water transport due to vessel collapse, leading to decreased turgor in the *exi* mutant cells; this negatively influences cell expansion, thus explaining their small size.

### 2.4. Leaf Morphology Mutants with Defects in Auxin Homeostasis

Many aspects of plant development rely on the phytohormone auxin, which promotes cellular differentiation and enhances or inhibits growth depending on the cellular context and its concentration. Auxin responses lead to changes in the transcriptome, but these responses can also occur rapidly and directly at the cell surface, although the underlying molecular mechanism is not completely understood (reviewed in [104,105]). Ten of the mutants (~5%) identified in our screens are defective in auxin homeostasis (Appendix A).

The *incurvata13* (*icu13*) mutant carries a C → T transition that generates a new splicing donor site in *AUXIN RESISTANT6* (*AXR6*), which encodes the CULLIN1 (CUL1) component of the Skip–Cullin–F-box TIR1 (SCF^TIR1^) ubiquitin ligase complex [106]. The *icu13* plants have a pleiotropic phenotype including diminished auxin responsiveness and leaves that are hyponastic, with a simple venation pattern, small adaxial pavement cells, densely packed palisade, and spongy mesophyll cells, as well as increased numbers of rosette leaves at bolting, early flowering, and reduced plant stature and apical dominance (Figure 4C and Appendix A; ref. [107]). We isolated the *icu6* mutant in our EMS screen and found that it carries a semi-dominant gain-of-function allele of *AXR3* (also named *INDOLE-3-ACETIC ACID INDUCIBLE17* [*IAA17*]), which encodes a transcriptional regulator that represses auxin-inducible gene expression. The heterozygous *icu6/AXR3* plants also have hyponastic leaves and smaller adaxial pavement cells but do not share any other phenotypes with *icu13*, as *icu6*/*AXR3* plants have smaller rosettes, pronounced apical dominance, and enhanced responsiveness to auxin (Figure 4E and Appendix A; ref. [108,109,110,111]). The *icu5* mutant turned out to carry a hypermorphic allele of *IAA3* (also named *SHORT HYPOCOTYL2* [*SHY2*]), encoding another repressor of auxin responses; we renamed *icu5* as *shy2-10* accordingly [107,112,113]. Leaves of the *shy2-10* mutant are smaller than those of the wild type and have more free-ending veins (Figure 4B and Appendix A; ref. [107]).

The loss-of-function *rotunda3* (*ron3*) mutant allele of *ROTUNDA3* (*RON3*) is characterized by shorter rosette lamina, smaller leaf width and area, together with a late-flowering phenotype, shorter roots exhibiting hypergravitropic growth, and lower apical dominance (Figure 4G and Appendix A; ref. [114]). A role for *RON3* in splicing and microRNA (miRNA) biogenesis was previously reported through the analysis of its *sickle-1* (*sic-1*) allele [115]. A Tandem Affinity Purification (TAP) assay revealed that RON3 interacts with six subunits of the cytosolic Protein Phosphatase 2A (PP2A) complex, which is involved in the polar localization of the PIN-FORMED (PIN) family of auxin efflux carriers [114,116,117,118]. In situ immunodetection of PIN1 and PIN2 in the root cells of the *ron3* mutant revealed a role for RON3 in PIN trafficking, promoting both basal internalization and apical delivery or maintenance of PIN auxin carriers [114]. Another physical interactor of RON3 is DEBRANCHING ENZYME1 (DBR1), which enables debranching of the lariat intronic RNAs produced by intron excision during splicing [119]. RON3 also represses photomorphogenic growth through its interaction with ELONGATED HYPOCOTYL 5 (HY5) and PHYTOCHROME INTERACTING FACTOR 4 (PIF4), two antagonistic transcription factors that specifically regulate growth largely under light [120].

Other genes under study in our laboratory are not directly related to auxin signaling, but their loss of function leads to altered auxin responses. The *ron1-1* mutant exhibits rounded leaves, with an open venation pattern, fewer lateral roots, late flowering, and loss of apical dominance (Figure 4F and Appendix A). *RON1* was previously reported as *FIERY1* (*FRY1*; [121]), *SAL1* [122], *HIGH EXPRESSION OF OSMOTICALLY RESPONSIVE GENES2* (*HOS2*; ref. [123]), and *ALTERED EXPRESSION OF APX2* (*ALX8*; ref. [124]). Although *RON1* encodes an enzyme with inositol polyphosphate 1-phosphatase and 3′(2′),5′-bisphosphate nucleotidase activities, which are required for inositol triphosphate (InsP_3_) metabolism, the venation pattern phenotypes of the *ron1-1* mutant suggest a role in auxin perception, as this phytohormone has a prevalent role in leaf vascular patterning [125,126]. Indeed, an interplay between inositol and auxin-mediated development has been reported, but has not been demonstrated yet [127,128].

Another example of a mutation that is not directly involved in auxin signaling but does cause an altered auxin response is *transcurvata1-1* (*tcu1-1*), an allele of *NUCLEOPORIN58* (*NUP58*). NUP58 is a component of the central barrier of the nuclear pore complex, along with NUP54 and NUP62, and thus, it is involved in nucleocytoplasmic trafficking of macromolecules [129,130]. The *tcu1-1* mutant has a pleiotropic phenotype with longer hypocotyls and petioles, reticulated, smaller, and asymmetrically epinastic leaf lamina, and early flowering (Figure 4H and Appendix A). Other *nup* mutants also show an early flowering phenotype, such as *nup54*, *nup62*, *nup96*, *nup98a nup98b*, *nup136*, and *nup160* [131,132,133,134,135,136]. Double mutant combinations (Appendix A) indicated a functional relationship between *TCU1* and *ICU5*, *AXR1* and *AXR3*, suggesting that TCU1 has a role in auxin signaling [130]. Indeed, AXR1 and AXR3 are partially retained in the cytoplasm of *nup54 nup58 nup62* triple mutant protoplasts [137]. The early flowering phenotype of the *tcu1-1* mutant is caused by the retention of the MADS-box transcription factor FLOWERING LOCUS C (FLC) in the cytoplasm [137].

**Figure 4 ijms-26-08332-f004:**
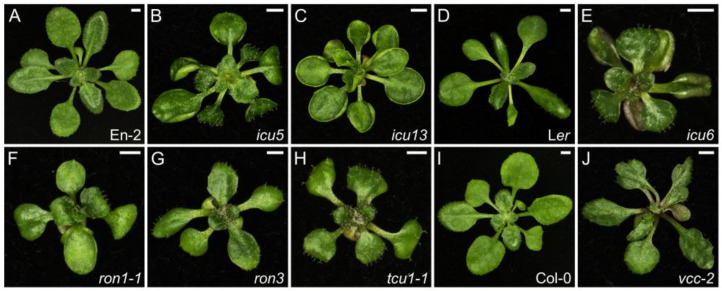
Leaf phenotypes of mutants with defects in auxin homeostasis. Rosettes of the wild-types Enkheim-2 (En-2) (**A**), L*er* (**D**), and Col-0 (**I**), and the *icu5* (*shy2-10*) (**B**), *icu13* (**C**), *icu6* (**E**), *ron1-1* (**F**), *ron3* (**G**), *tcu1-1* (**H**), and *deal1-1* (*vcc-2*) (**J**) homozygous mutants. Photographs were taken 21 das. Scale bars, 2 mm. These mutants were described in [107] (**B**,**C**), ref. [111] (**E**), ref. [125] (**F**), ref. [114] (**G**), ref. [130] (**H**), and [138] (**J**).

In our PhenoLeaf collection (https://genetics.umh.es/query-phenoleaf-db/ (accessed on 1 July 2025); ref. [26]), the *desigual1-1* (*deal1-1*) mutant line stood out because of its unusual phenotype, as its leaves clearly deviated from the bilateral symmetry normally seen in the wild type, and without visible alterations in dorsoventral polarity [138]. We later identified *DEAL1* as the previously reported gene *VASCULATURE COMPLEXITY AND CONNECTIVITY* (*VCC*; ref. [139]), which prompted us to rename the *deal1* mutants as *vcc*. The leaf margins of *vcc* plants show aberrant lobes and sinuses that are bigger than those of the wild type, whose randomness ultimately causes loss of bilateral symmetry (Figure 4J and Appendix A). Although alterations to the leaf medio-lateral axis are usually caused by changes in dorsoventral polarity [140], *vcc* mutants only show alterations in the medio-lateral axis without disturbing either the leaf dorsoventral axis or the proximo-distal axis. Notably, these differences in growth are related to cell proliferation, not cell expansion. The auxin maxima are unevenly located in the leaf primordia of *vcc* mutants, which do not show alterations in the venation pattern. CUP-SHAPED COTYLEDON2 (CUC2), a transcriptional regulator that influences auxin distribution, is expressed in well-defined and evenly spaced domains in leaf primordia of the Columbia-0 (Col-0) wild type. This pattern is lost in the *vcc* mutants, which show abnormally close or distant consecutive CUC2 domains, whose disposition was asymmetrical in primordia margins. VCC appears to be necessary for communication between growth-promoting and growth-repressing signals by mediating auxin and CUC2 signaling [138].

Our mutants with defects in auxin signaling are difficult to classify due to the diverse phenotypes arising from the complexity of auxin responses, which can alter the transcriptome in vastly different ways depending on the mutated gene. In fact, some mutants exhibit opposite phenotypes, such as lower or greater auxin responsiveness, early or late flowering, more complex or simplified venation patterns, and hyponastic or epinastic leaves with rounded or aberrant lobes and sinuses. This diversity can be explained because auxin response depends on both the correct auxin trafficking mediated by PIN proteins and apparently modulated by RON3, RON1, and VCC; and the proper SCF-mediated ubiquitination of IAA proteins followed by 26S proteasome degradation, which is presumably impaired in *icu13*, *icu6*, *icu5*, and *tcu1* mutants. Given the high number of leaf morphology mutants that exhibit altered auxin responses (Appendix A), the groups of Karin Ljung and Ondřej Novák developed a high-throughput method to screen for altered auxin metabolite profiles, which they tested using our collection of mutants, and which should be valuable for other large-scale mutant collections [141].

### 2.5. Leaf Morphology Mutants with Impaired miRNA Biogenesis and Function

An important mechanism of post-transcriptional regulation of gene expression is accomplished via endogenous small RNA molecules known as miRNAs, which impose translation inhibition, mRNA cleavage, and/or RNA-directed DNA methylation depending on the target transcript or gene (reviewed in [142]). At least eight mutants (~4%) from our screens are related to miRNA biogenesis (Appendix A).

In addition to leaf upward curvature, the *incurvata* mutants *icu8*, *icu9*, and *icu15* share additional leaf phenotypes, including a poorly defined boundary between lamina and petiole and the presence of trichomes on the abaxial epidermis, fewer stomata per leaf, and a simpler venation pattern than in the wild type (Figure 5B–F and Appendix A). These *icu* mutants are late flowering, have small and compact inflorescences with prematurely open flowers, incompletely fused carpels, and reduced fertility. The phenotypic similarities among these mutants suggested that they function in the same developmental program, and in fact all have been shown to participate in miRNA biogenesis and function. Indeed, *icu8*, *icu9*, and *icu15* were shown in the laboratory of María Rosa Ponce to carry new alleles of the well-characterized genes *HYPONASTIC LEAVES1* (*HYL1*), *ARGONAUTE1* (*AGO1*), and *HUA ENHANCER1* (*HEN1*), respectively [143]. HYL1 acts in several steps of miRNA biogenesis, from the transcriptional regulation of *MIR* genes to the inhibition of target mRNA translation, and has other roles not related to miRNA signaling such as the transcriptional regulation of genes related to plastid organization (reviewed in [144]). HEN1 methylates the miRNA/miRNA* duplex at the 3′ terminal nucleotides [145] and interacts with HYL1 [146]. AGO1 is the catalytic subunit of the RNA-induced silencing complex (RISC), the effector complex that mediates gene silencing [147].

The *TCU2* gene, which encodes the auxiliary subunit of the NatB N-alpha-acetyltransferase complex [148], was serendipitously found to interact with *AGO10*. Our *tcu2-1* mutant was isolated in the Landsberg *erecta* (L*er*) background and showed a stronger phenotype than the *tcu2-2* mutant in the Col-0 background, suggesting the presence of a genetic modifier in L*er*. This genetic modifier was *zwille-2* (*zll-2*) [149], a previously described allele of *AGO10*, and the *tcu2-1 zll-2* double mutant showed a synergistic leaf phenotype with strong folding of the leaf (Appendix A), which suggests a functional relationship between *TCU2* and *AGO10*. Otherwise, the loss of *TCU2* function alone suggest some degree of auxin deficiency, since the *tcu2* mutants show leaf reticulation and downward folding at an oblique angle relative to the primary vein, early flowering, unfertilized or aborted ovules in siliques, and a greater proportion of siliques with more than two valves than in the L*er* and Col-0 wild-type accessions (Figure 5G and Appendix A).

Our mutants with perturbed silencing by miRNAs, like those in auxin signaling, have no easily predictable effects on the overall plant phenotype. Depending on the specific mutated component of the miRNA pathway, the resulting phenotype can vary substantially, even displaying opposite traits, as seen with the *icu* and *tcu2* mutants. The *icu* mutants have hyponastic leaves and flower late, whereas the *tcu2* mutants have epinastic leaves and early flowering. This variability in phenotypic outcomes presents challenges when identifying mutants with altered miRNA biogenesis based solely on plant morphology but evidences the need for spatial and temporal gene expression fine-tuning for proper morphogenesis.

### 2.6. Leaf Morphology Mutants with Altered Epigenetic Machinery

In plants, acclimation, vernalization, systemic acquired resistance, and imprinting are mitotically heritable and epigenetically regulated [150,151]. Other aspects of plant development and physiology that have been considered to be at least partially under epigenetic control include morphological plasticity, light signaling, stress responses, male and female gametogenesis, and flowering time. A recent review of the epigenetic machinery in plants highlights the revolutionary potential of advanced epigenetic tools to enhance crop productivity [152].

Polycomb group (PcG) proteins are key components of the epigenetic machinery, are highly conserved among eukaryotes, and epigenetically repress the expression of many genes, including those controlling growth, development, and environmental adaptation [153,154]. PcG proteins form part of two heteromultimeric Polycomb Repressive Complexes (PRCs) with different epigenetic functions: PRC1 is a ubiquitin ligase targeting histone H2A, whereas PRC2 is an H3K27 methyltransferase [155]. Seventeen mutants (~8%) of our mutant collection carry mutations in genes belonging to the epigenetic machinery (Appendix A).

The proteins encoded by seven genes identified from our mutant screens and/or studied prior to 2009 have epigenetic functions: CURLY LEAF [156] (CLF, which we initially called INCURVATA1 [ICU1] [157]), a PRC2 core component; the histone acetyltransferases ELONGATA1–4 (ELO1–ELO4) [158,159]; HISTONE MONOUBIQUITINATION1 (HUB1, which we initially named ANGUSTA4 [ANG4]), an E3 ligase that monoubiquitinates H2B [160]; and INCURVATA2 (ICU2), the catalytic subunit of DNA polymerase α, which interacts with chromatin remodeler complexes and whose loss of function affects the maintenance of repressive chromatin states at loci that are PRC2 targets [157,161,162].

The *icu11-1* mutant has a pleiotropic phenotype that includes hyponastic leaves, early flowering, smaller palisade mesophyll cells, and lower fertility than the wild type (Figure 6 and Appendix A). *ICU11* encodes a 2-oxoglutarate and Fe(II)-dependent dioxygenase [163] that appears to be a PRC2 accessory protein that facilitates demethylation of the H3K36me3 active mark [163,164]. ICU11 and its closest paralog, CUPULIFORMIS2 (CP2), show unequal functional redundancy and are required for vegetative development in Arabidopsis, as *icu11 cp2* double mutant plants skip the vegetative phase and develop floral organs immediately upon germination (Appendix A; ref. [163,165]). CP2 may also act as a PRC2 accessory protein, and a transcriptome landscape of the *icu11-5 cp2-1* double mutant resembles that of the loss-of-function PRC2 mutant *embryonic flower 2-3* (*emf2-3*), which lacks EMF2, a core PRC2 component [166].

Comparing the phenotypes of the *icu11-1* mutant and epigenetic signaling mutants previously studied in our laboratory [160,161] does not reveal a distinct characteristic phenotype for this class of mutants. As mentioned in previous sections, this may be due to the broad influence of these mutations on plant functions that affect the expression of many genes. Indeed, the phenotype of these mutants can be explained by the heterochronic and ectopic derepression of genes. For example, the *clf*, *icu2*, and *icu11* mutants show an early flowering phenotype caused by the expression of floral homeotic genes such as *AGAMOUS* or *SEPALLATA3* [156,161,163].

## 3. Concluding Remarks

From our mutant screens, we identified 208 mutants affecting the morphology of Arabidopsis leaves, which we found to fall into 144 complementation groups (Appendix A). This set of mutants does not include the insertional mutants that we selected from the Arabidopsis SALK collection and described in the PhenoLeaf database, which comprises 706 mutant lines [26]. Of these 208 mutants, 83 were found to carry alleles of 41 previously characterized genes, and characterization of 30 additional mutants identified 22 genes not previously studied at the functional level (Appendix A). In the present review, we have summarized the findings obtained from the analysis of 52 mutants (corresponding to 38 genes) since 2009, in some cases in collaboration with other groups or by other groups. We are currently studying 13 *DEN* genes, but we still have 74 mutant lines yet to be studied, which presumably correspond to 61 different genes (Appendix A). Further analysis of these mutants will surely unveil previously missed functions related to Arabidopsis leaf development.

Some functional classes have conspicuous phenotypes. For example, all mutants related to protein translation mentioned above have either narrow or toothed leaves (Figure 1), whereas most mutants with defects in auxin signaling show defects in their leaf venation pattern (Figure 4). Loss of chloroplast function can lead to chlorotic or reticulated phenotypes, associated with a lower accumulation of chlorophyll *a* and/or *b*; this relationship is rather obvious, but it can also produce unexpected effects such as irregular leaf margins with prominent teeth, small laminae, or necrotic patches within leaves (Figure 2).

Connections between two apparently unrelated biological programs were revealed when the relationship between protein function and phenotype was not as straightforward. This was the case with *api7*, an allele of *ABCE2*, which encodes a protein involved in ribosome dissociation, but the simpler venation pattern of the *api7* mutant suggested an alteration in auxin signaling, as was previously proposed for the *ABCE2* ortholog of *Cardamine hirsuta* [58,167]. This is also the case for *ven4-0*, as VEN4 is primarily involved in maintaining the dNTP pool. Loss of VEN4 function disrupts chloroplast development, because it is highly dependent on the fine-tuning of dNTP levels [91,168]. The late-flowering phenotype of *rug1* [78] presumably reflects a defect in chloroplast-to-nucleus retrograde signaling. The loss of function of RUG1, which is involved in RNA editing, or the loss of the central regulator of retrograde signaling GUN1 leads to a late-flowering phenotype [78,169]. Furthermore, RNA editing of chloroplast transcripts and retrograde signaling have a complex regulatory relationship [80]. Another example is the InsP_3_ metabolic enzyme RON1, which shows a phenotype reminiscent of mutants defective in auxin signaling, consistent with the interplay previously described for those pathways [125,127,128].

Other mutants encoding components of regulatory pathways that affect multiple developmental programs have pleiotropic phenotypes inherently unpredictable prior to empirical observation. For example, *icu11-1* has an early flowering phenotype caused by the ectopic and heterochronic derepression of genes encoding MADS-box transcription factors that promote flowering, such as *AGAMOUS* (*AG*), *SEPALLATA1* (*SEP1*), *SEP2*, or *SEP3*, but it also has higher expression levels of the flowering repressor *FLC* [163,164]. Therefore, given that ICU11 influences the expression of opposite components of the same pathway, it is not easy to predict what evokes the early flowering phenotype of *icu11* mutants. This unpredictability is reflected in the disparity of phenotypic traits among the miRNA biogenesis and epigenetic mutants mentioned in this review.

It is of note that our forward genetic screens resulted in the first isolation and characterization of viable mutant alleles of the *SECA2*, *NAP14*, *ANU7*, *VEN3*, and *VEN6* genes, and the first mutant alleles of *ANU10* and *RUG1*. In addition, the *rug2* mutation caused phenotypes different from that of its previously characterized allele *bms*, which is embryonic lethal.

There are still open questions regarding the mutants discussed in this review. We still do not know the molecular functions of *ANU7*, *ANU10*, or *VEN5*, nor do we understand the nature of the genetic interaction between *TCU1* and *AGO10*. The main question that remains is the one that prompted these screens for leaf morphology mutants: how does a leaf form? Although these screens have been fruitful, and the analysis of the above mutants allowed us to make important contributions to the field, the genetics of leaf morphogenesis still needs many more studies to fully elucidate this complex developmental program.

Looking ahead, advances in automated image analysis and/or machine learning-based phenomics—such as those described in [170,171,172]—may transform how Arabidopsis leaf mutants are characterized, enabling cross-scale analyses from individual cells to whole rosettes. Although these methods still require rigorous benchmarking and community standards, their integration with classical genetic approaches has the potential to reshape future mutant screens and broaden our understanding of leaf development.

## Figures and Tables

**Figure 5 ijms-26-08332-f005:**
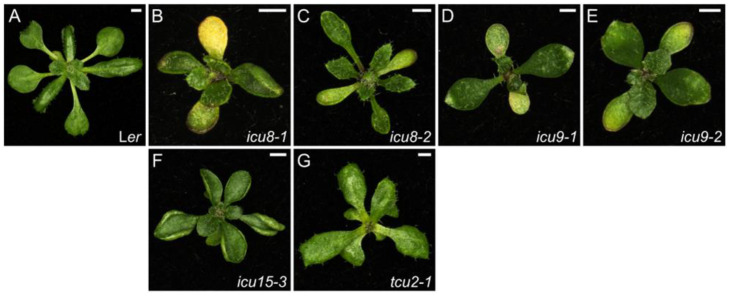
Leaf phenotypes of mutants altered in miRNA biogenesis and function. Rosettes of the wild-type L*er* (**A**), and the *icu8-1* (*hyl1-11*) (**B**), *icu8-2* (*hyl1-12*) (**C**), *icu9-1* (*ago1-51*) (**D**), *icu9-2* (*ago1-52*) (**E**), *icu15-3* (*hen1-13*) (**F**), and *tcu2-1* (**G**) homozygous mutants. Photographs were taken 21 das. Scale bars, 2 mm. These mutants were described in [143] (**B**–**F**) and [148] (**G**).

**Figure 6 ijms-26-08332-f006:**
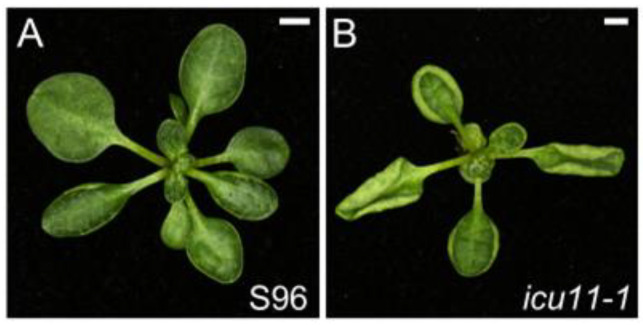
Leaf phenotype of *icu11-1*, a mutant altered in the epigenetic machinery. Rosettes of the wild-type S96 (**A**) and the *icu11-1* homozygous mutant (**B**). Photographs were taken 21 das. Scale bars, 2 mm. This mutant was described in [163] (**B**).

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
