# Peer review of "Light and Shadows: Insights from Large-Scale Visual Screens for Arabidopsis Leaf Morphology Mutants"

_ijms, 2025, doi:10.3390/ijms26178332_

Round 1

Reviewer 1 Report

Comments and Suggestions for Authors

In the article entitled "Light and Shadows: Insights from Large-Scale Visual Screens for Arabidopsis Leaf Morphology Mutants ," the authors revisit a 2009 study that classified Arabidopsis mutants based on how disrupted pathways affected leaf morphology. This approach is both innovative and promising, offering potentially significant insights for plant research.

To improve the manuscript, the authors might consider simplifying the description of the different mutants and the pathways they impact, perhaps including a diagram for clarity.

Author Response

Comments 1: To improve the manuscript, the authors might consider simplifying the description of the different mutants and the pathways they impact, perhaps including a diagram for clarity.

Response 1: We appreciate the reviewer’s suggestion. However, we believe that retaining detailed descriptions of the phenotypes is important for the usefulness of this review, as such information can help researchers recognize which developmental processes may be affected in their own mutants of interest. Regarding the idea of adding a diagram, we carefully explored possible formats but did not find a straightforward way to represent the complexity of the data without oversimplifying or risking confusion. For these reasons, we opted to maintain the current structure, which we consider the clearest way to present the information.

Reviewer 2 Report

Comments and Suggestions for Authors

Review of Manuscript: «Light and Shadows: Insights from Large-scale Visual Screens for Arabidopsis Leaf Morphology Mutants».

The manuscript under review presents a comprehensive overview of results obtained from large-scale forward genetic screens and leaf morphology mutants for higher plants, in particular Arabidopsis thaliana. The authors summarize and functionally reinterpret a substantial mutant collection, revisiting previously described phenotypes and expanding upon them with new functional categorization. The review covers a wide range of biological processes involved in leaf development, including translation, chloroplast biogenesis, cell wall biosynthesis, auxin homeostasis, microRNA pathways, and epigenetic regulation. The manuscript is generally well-structured, informative, and provides useful insights into the genetic architecture of leaf morphogenesis.

Below is a detailed list of my comments and observations regarding the manuscript.

One notable point is that the abstract and certain parts of the main text are written in the first person ("we present", "we found"), which is not conventional in review articles and should be revised to a more neutral, third-person tone.

Figures are well-prepared and support the text. While figures are helpful, summary tables containing gene identities, phenotypes, and references would be more informative and appropriate for a review article. I suggest integrating Table S1 into the main text.

This statement requires a proper reference: The first plant mutant affecting a gene encoding a ribosomal protein (RPS18A) was Arabidopsis pointed first leaves (pfl), which was isolated in the laboratory of Mieke Van Lijsebettens.

I recommend reviewing the entire manuscript for consistency in formatting gene names, alleles, and proteins, in accordance with Arabidopsis nomenclature guidelines. (anu10-1 and ANU10 and so on throughout the text).

Although the main focus of the review is Arabidopsis thaliana, several other model organisms (e.g., Drosophila melanogaster, Oryza sativa) are discussed in the manuscript. Consider clarifying this in the title or limiting the scope more strictly to Arabidopsis.

I propose to remove the brackets in this sentence and immediately introduce the names thylakoids, grana and stroma, which are well-established concepts in plant physiology. «The chloroplast consists of membranous disks (thylakoids), organized in stacks (grana) and surrounded by fluid (stroma), all enclosed by two membrane layers.»

The secretory pathway SECA2 protein and TOC33 participate in the import of proteins from the cytosol into chloroplasts [66,67]. And SECA2 and TOC33 are proteins in this context?

If the authors refer to their PhenoLeaf database, it is better to provide a reference everywhere [26].

On page 15, in the sentence “This set of mutants does not include the insertional mutants that we selected from the Arabidopsis SALK collection and described in the PhenoLeaf database, which comprises 708 mutant lines (Wilson-Sánchez et al., 2014),” the reference to the article should be brought to the general form of numbering.

The manuscript is scientifically sound and presents a valuable synthesis of a significant genetic resource. Nevertheless, revisions are necessary, particularly regarding formatting consistency, nomenclature, and textual clarity, before it can be accepted for publication.

Author Response

Comments 1: One notable point is that the abstract and certain parts of the main text are written in the first person ("we present", "we found"), which is not conventional in review articles and should be revised to a more neutral, third-person tone.

Response 1: Thank you for this observation. Because this review centers on a mutant collection generated in our laboratory, we felt that use of the first person was the clearest way to convey our direct involvement in the work.

Comments 2: Figures are well-prepared and support the text. While figures are helpful, summary tables containing gene identities, phenotypes, and references would be more informative and appropriate for a review article. I suggest integrating Table S1 into the main text.

Response 2: We appreciate this helpful suggestion and fully understand its motivation. However, we believe it is preferable to keep Table S1 as supplementary material. The table is extensive, and incorporating it into the main text would considerably increase the length of an already long manuscript. More importantly, moving it to the main text would require renumbering a large number of references: in the current version, each reference is cited at the point where a given mutant is first mentioned, but if the table were inserted into the main text, all references would instead be renumbered according to the table’s position. We feel that this would disrupt the logical flow of the manuscript and make it harder for readers to follow. For these reasons, we have opted to retain Table S1 as supplementary material, while ensuring that it is clearly cross-referenced in the main text for accessibility.

Comments 3: This statement requires a proper reference: The first plant mutant affecting a gene encoding a ribosomal protein (RPS18A) was Arabidopsis pointed first leaves (pfl), which was isolated in the laboratory of Mieke Van Lijsebettens.

Response 3: Thank you for pointing this out. We have now added reference [53] directly to the relevant sentence to ensure proper attribution (page 3).

Comments 4: I recommend reviewing the entire manuscript for consistency in formatting gene names, alleles, and proteins, in accordance with Arabidopsis nomenclature guidelines. (anu10-1 and ANU10 and so on throughout the text).

Response 4: We appreciate this suggestion, also raised by Reviewer 4. We carefully re-examined the entire manuscript for inconsistencies in nomenclature. Throughout the text, proteins are written in uppercase (e.g., ANU10), genes in uppercase italics (ANU10), and mutants, mutations, and mutant alleles in lowercase italics (anu10-1). We did not detect formatting errors, but we have double-checked to ensure full compliance with Arabidopsis nomenclature guidelines (Meinke and Koornneef, 1997).

Comments 5: Although the main focus of the review is Arabidopsis thaliana, several other model organisms (e.g., Drosophila melanogasterOryza sativa) are discussed in the manuscript. Consider clarifying this in the title or limiting the scope more strictly to Arabidopsis.

Response 5: We appreciate this point. The review is indeed focused on Arabidopsis thaliana, as reflected in the title. References to other organisms are limited to the first sentences of the introduction and serve only to provide context.

Comments 6: I propose to remove the brackets in this sentence and immediately introduce the names thylakoids, grana and stroma, which are well-established concepts in plant physiology. «The chloroplast consists of membranous disks (thylakoids), organized in stacks (grana) and surrounded by fluid (stroma), all enclosed by two membrane layers.»

Response 6: Thank you for the suggestion. We have revised the sentence as recommended, introducing “thylakoids”, “grana,” and “stroma” directly without brackets (page 5, paragraph 4, lines 1 and 2).

Comments 7: The secretory pathway SECA2 protein and TOC33 participate in the import of proteins from the cytosol into chloroplasts [66,67]. And SECA2 and TOC33 are proteins in this context?

Response 7: Yes, SECA2 and TOC33 are indeed proteins involved in the import of proteins into chloroplasts. We have clarified this in the text to avoid confusion (page 5).

Comments 8: If the authors refer to their PhenoLeaf database, it is better to provide a reference everywhere [26].

Response 8: We acknowledge this point and have added reference [26] to every mention of the PhenoLeaf database (pages 3 and 15).

Comments 9: On page 15, in the sentence “This set of mutants does not include the insertional mutants that we selected from the Arabidopsis SALK collection and described in the PhenoLeaf database, which comprises 708 mutant lines (Wilson-Sánchez et al., 2014),” the reference to the article should be brought to the general form of numbering.

Response 9: Thank you for noticing this. We have corrected the reference format on page 15, ensuring consistency with the numbering system used throughout the manuscript.

Comments 10: The manuscript is scientifically sound and presents a valuable synthesis of a significant genetic resource. Nevertheless, revisions are necessary, particularly regarding formatting consistency, nomenclature, and textual clarity, before it can be accepted for publication.

Response 10: We appreciate the reviewer’s overall positive evaluation of the manuscript. Regarding the request for further revisions to formatting, nomenclature, and textual clarity, we have carefully re-checked the entire text and tables for consistency and readability. If the reviewer has specific instances in mind that may have escaped our attention, we would be grateful to receive them, and we will gladly make the corresponding corrections.

Reviewer 3 Report

Comments and Suggestions for Authors

The article is devoted to a large-scale visual screening of Arabidopsis thaliana mutants with altered leaf shape. The goal of the work is to understand the genetic causes of these differences. 

The topic is relevant, as interest in high-throughput phenotyping and plant developmental genetics is growing. Screening of such a scale is rare. The authors combined classical mutagenesis and modern methods of data collection and analysis. The work expands the list of known leaf morphotypes and highlights possible genes for further research. 

I think that the text would have benefited from discussing automated visualization and quantitative indices of shape, as well as the influence of environmental conditions. 

The conclusions are generally consistent with the presented data. The authors acknowledge limitations, including the subjectivity of visual assessment. However, in some cases, the relationship between mutant groups and putative genetic pathways is preliminary and requires experimental confirmation, this would have benefited from more explicit discussion in the text. 

The list of references is well chosen and covers key works on arabidopsis leaf development. Additionally, it would be possible to refer to new research in the field of phenotyping using machine learning. 

The figures are informative and help to understand the classification of mutants. For greater usefulness, it would be worth adding high-resolution images to the supplementary materials or creating a separate database. 

Author Response

Comments 1: I think that the text would have benefited from discussing automated visualization and quantitative indices of shape, as well as the influence of environmental conditions. 

Response 1: We appreciate this insightful comment. Automated visualization and quantitative shape indices, as well as environmental influences on leaf morphology, are indeed important and rapidly advancing areas. However, a detailed discussion of these topics lies beyond the scope of the present review, which focuses on classical large-scale mutant screening. We agree, nonetheless, that these approaches will be valuable for future studies, and we will consider them in forthcoming work. A paragraph on this has been added to the end of the discussion section of the manuscript (page 16, paragraph 3).

Comments 2: The conclusions are generally consistent with the presented data. The authors acknowledge limitations, including the subjectivity of visual assessment. However, in some cases, the relationship between mutant groups and putative genetic pathways is preliminary and requires experimental confirmation, this would have benefited from more explicit discussion in the text. 

Response 2: We thank the reviewer for this valuable observation. We have revised the text to clarify which proposed connections between certain mutant classes and genetic pathways remain hypothetical and still require experimental validation. Although we already had consistently used expressions such as “may have” or “suggest”, we have now strengthened this point by explicitly adding in two instances the statements “but this has not yet been experimentally confirmed” and “but has not been demonstrated”. We believe this addition strengthens the balance between established knowledge and open questions in the field.

Comments 3: The list of references is well chosen and covers key works on arabidopsis leaf development. Additionally, it would be possible to refer to new research in the field of phenotyping using machine learning. 

Response 3: We appreciate this suggestion. We have now added three citations to recent studies on plant phenotyping, thereby complementing the classical approaches highlighted in the review to the new paragraph that we mention in our Response 1 to Reviewer 3 (page 16).

Comments 4: The figures are informative and help to understand the classification of mutants. For greater usefulness, it would be worth adding high-resolution images to the supplementary materials or creating a separate database. 

Response 4: We thank the reviewer for this excellent suggestion. The idea of compiling a database with high-resolution images of all mutants obtained in our screenings is indeed valuable and would greatly benefit the community. Unfortunately, creating such a resource is not feasible within the limited timeframe available for the current revision. Nevertheless, we plan to explore this possibility in the future as an extension of our PhenoLeaf database.

Reviewer 4 Report

Comments and Suggestions for Authors

I go through the manuscript and found it very interesting in comparatively analyzing the leaf phenotypes of Arabidopsis. The manuscript is well written however require a proof reading to correct minor typos such as please rearrange the keywords into ascending alphabetic order. Further there some genes names such as ferredoxin etc which needs to be italic and protein names should be kept straight through out the manuscript. I accept the manuscript for publication. 

Author Response

Comments 1: The manuscript is well written however require a proof reading to correct minor typos such as please rearrange the keywords into ascending alphabetic order.

Response 1: Thank you for noticing this detail. We have proofread the manuscript again and corrected minor issues, including rearranging the keywords in alphabetical order as suggested.

Comments 2: Further there some genes names such as ferredoxin etc which needs to be italic and protein names should be kept straight through out the manuscript.

Response 2: We appreciate this comment, which was also raised by Reviewer 2. We have carefully re-examined the entire manuscript for consistency in the formatting of gene, protein, and mutant names. In line with Arabidopsis nomenclature guidelines, genes are written in uppercase italics, proteins in uppercase roman letters, and mutants, mutations, and mutant alleles in lowercase italics. In the case mentioned, “ferredoxin” refers to a protein and is therefore correctly not italicized. Nonetheless, we have re-checked the text thoroughly to ensure that no formatting inconsistencies remain.